# Efficient Identification of Direct Causal Parents via Invariance and Minimum Error Testing

**Minh Nguyen**                                                                          *bn244@cornell.edu*
*School of Electrical and Computer Engineering, Cornell University and Cornell Tech*

**Mert R. Sabuncu**                                                                      *msabuncu@cornell.edu*
*School of Electrical and Computer Engineering, Cornell University and Cornell Tech*
*Department of Radiology, Weill Cornell Medicine*

**Reviewed on OpenReview:** *https://openreview.net/forum?id=3G7mFdGVRW*

## Abstract

Invariant causal prediction (ICP) is a popular technique for finding causal parents (direct causes) of a target via exploiting distribution shifts and invariance testing (Peters et al., 2016). However, since ICP needs to run an exponential number of tests and fails to identify parents when distribution shifts only affect a few variables, applying ICP to practical large scale problems is challenging. We propose MMSE-ICP and fastICP, two approaches which employ an error inequality to address the identifiability problem of ICP. The inequality states that the minimum prediction error of the predictor using causal parents is the smallest among all predictors which do not use descendants. fastICP is an efficient approximation tailored for large problems as it exploits the inequality and a heuristic to run fewer tests. MMSE-ICP and fastICP not only outperform competitive baselines in many simulations but also achieve state-of-the-art result on a large scale real data benchmark.

## 1 Introduction

Causal discovery (CD) can offer insights into systems' dynamics (Pearl, 2018) which are helpful for influencing some outcomes (e.g. having diseases or not) or creating robust ML models. For example, a model based on a target's causal parents can robustly predict the target despite various distribution shifts. CD can be global or local: global CD searches for the complete causal graph while local CD only searches for causal relations surrounding a specific target $Y$. Thus, global CD is often intractable as the search domain grows exponentially with the number of variables. Local CD is more tractable and is sufficient if the goal is to change $Y$ through interventions or to build domain-invariant ML models of $Y$. Recently, many works have built on local CD to improve ML models' out-of-distribution generalization (Rojas-Carulla et al., 2018; Magliacane et al., 2018; Arjovsky et al., 2019; Christiansen et al., 2021).

Invariant causal prediction (ICP) is a local CD method that finds the causal parents of $Y$ using the invariance property: the distribution of $Y$ conditioned on all of its causal parents will be invariant under interventions on variables other than $Y$ (Peters et al., 2016). Specifically, ICP tests all subsets of variables for invariance and outputs the intersection (denoted as $\hat{\mathbf{S}}_{\text{ICP}}$) of all invariant subsets. As the number of subsets grows exponentially, applying ICP to large problems (> 20 variables) is difficult. Besides, ICP is guaranteed to identify all causal parents only when every variable in the system except $Y$ is perturbed (Peters et al., 2016). However, in many practical problems (e.g. problems in cell biology with 20,000 variables), it is almost impossible to perturb all variables (Uhler, 2024). With limited perturbation/intervention, ICP may fail to identify causal parents (Rosenfeld et al., 2021; Mogensen et al., 2022), as disjoint subsets can be invariant, yielding an empty intersection (see Figure 1).

Invariant ancestry search (IAS) addresses ICP's failure to identify causal parents by outputting the causal ancestors of $Y$ instead (Mogensen et al., 2022). The causal ancestors identified is the union (denoted

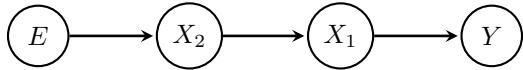

Figure 1: Consider this directed acyclic graph (DAG) representing a noisy causal structural model as a motivating example, where no causal mechanism is noise-free and thus there are no duplicate variables. Let $E$ denote the variable capturing the environment (or context) (Mooij et al., 2020). Its children are directly affected by intervention. This reflect distribution shifts between contexts. $Y$ is the target variable. $\hat{\mathbf{S}}_{\text{ICP}} = \emptyset$ because the invariant sets are $\{X_1\}$, $\{X_2\}$, and $\{X_1, X_2\}$. In contrast, $\hat{\mathbf{S}}_{\text{IAS}} = \{X_1, X_2\}$. Our methods output $\{X_1\}$ since the prediction error of $\hat{Y}_M(X_1)$ is less than $\hat{Y}_M(X_2)$'s.

as $\hat{\mathbf{S}}_{\text{IAS}}$) of all minimally invariant subsets. A set is minimally invariant if none of its proper subsets is invariant (Mogensen et al., 2022). Like ICP, $\hat{\mathbf{S}}_{\text{IAS}}$ can also be used to construct a stable predictor of $Y$ across environments even though it is not the most compact stable predictor. Besides, intervening on an ancestor may not be as effective in influencing the value of $Y$ as intervening on a parent. IAS can be sped up by only testing subsets up to a certain size at the cost of potentially finding fewer causal ancestors, thus trading accuracy for speed.

We propose two approaches that employ an error inequality to address the identifiability and scalability problem of ICP. The error inequality implies that the prediction error of the predictor using $Y$'s causal parents is the smallest among all predictors which do not use $Y$'s descendants. Thus, the first approach, MMSE-ICP (short for *minimum mean squared error ICP*), finds invariant sets that do not contain $Y$'s descendants and outputs the invariant set with the smallest error for predicting $Y$. The second approach is fastICP, an efficient approximation that relies on a heuristic and constrained search scheme (see Section 3.4). fastICP also exploits the error inequality to reduce the number of tests for invariance to identify the causal parents. Consequently, fastICP can handle large-scale problems with numerous variables. Experiments on simulated and real data show MMSE-ICP and fastICP outperforming competitive baselines.

## 2 Background

In different environments/settings, data are generated using different generative mechanisms. An intervention refers to perturbation of generative mechanism of a specific variable (Pearl, 2009). Observational setting indicates no perturbation. Interventional setting indicates one or more variables are perturbed. Experimental data belong to interventional setting in which the interventions are known (which variables are perturbed and how are they perturbed). Most naturally occurring distribution-shift data also fall under interventional setting, however the interventions are unknown.

Many CD methods use solely observational data. Constraint-based (Spirtes et al., 2000) and score-based (Chickering, 2002) methods can identify up to the Markov equivalent class (MEC) of the true graph, leaving some edges with unresolved direction. With additional assumptions (e.g. non-Gaussianity (Shimizu et al., 2006), nonlinearity (Hyvärinen & Pajunen, 1999; Hoyer et al., 2008; Zhang & Hyvärinen, 2009; Peters et al., 2014; Zhang et al., 2015; Rolland et al., 2022), or independent causal mechanisms (Janzing et al., 2012)), methods are able to resolve more edges (Glymour et al., 2019). Optimization-base methods (Zheng et al., 2018; Geffner et al.) have become popular recently despite the lack of identifiability guarantees (Mogensen et al., 2022).

Having data from multiple settings can improve identifiability. When interventions are known, undirected edges in the MEC first estimated using observational data can be resolved using interventional data (He & Geng, 2008). Other approaches jointly model interventional and observational data (Hauser & Bühlmann, 2012; 2015; Wang et al., 2017). There are also optimization-based methods that leverage both types of data (Lorch et al., 2021; Lippe et al., 2022). In contrast, UT-IGSP (Squires et al., 2020), SDI (Ke et al., 2023), and DCDI (Brouillard et al., 2020) are methods that can work with unknown interventions. ICP (Peters et al., 2016; Pfister et al., 2019; Gamella & Heinze-Deml, 2020; Martinet et al., 2022) and IAS (Mogensen et al., 2022) both assume unknown interventions and only depends on the invariance property to identify causal structures. Our approaches also assume unknown interventions.

Prototypical global CD methods such as PC (Colombo et al., 2014; Li et al., 2019) and GES (Chickering, 2002) can be slow, so faster global CD ones (Cheng et al., 2002; Tsamardinos et al., 2006; Ramsey et al., 2017) have been proposed, although they make strong assumptions that may be inappropriate in some problems. In contrast, local CD methods only identify local causal structures surrounding a target variable. For example, several algorithms only search for the Markov Blanket (MB) of the target (i.e. its parents, children, and spouses) (Koller et al., 1996; Margaritis & Thrun, 1999; Tsamardinos et al., 2003; Gao & Ji, 2015; Yang et al., 2021). Consequently, local CD methods often have faster runtime or can afford to make milder assumptions leading to more accurate identification. ICP and IAS only search for parents and ancestors of the target respectively, omitting all descendants of the target. They both make very mild assumptions and offer guarantees in terms of false positive rates. Another local CD approach is CORTH (Soleymani et al., 2022) which is based on double ML (Chernozhukov et al., 2018). CORTH assumes that $Y$ is not a parent of any other variable (a strong assumption which can be unrealistic) and that $Y$ is a linear combination of other variables, and achieves linear runtime finding parents of $Y$.

## 3 Method

### 3.1 Definitions and assumptions

We represent causal relations between variables using a Directed Acyclic Graph (DAG), where each node is a variable and directed edges between nodes represent direct causal influence (Pearl, 2009). We use the usual notations from graphical models (Lauritzen, 1996). Specifically, $\mathsf{PA}(Y)$, $\mathsf{CH}(Y)$, $\mathsf{AN}(Y)$, $\mathsf{DE}(Y)$, and $\mathsf{ND}(Y)$ denote the parents, children, ancestors, descendants, and non-descendants of $Y$ respectively. Thus, $\mathsf{PA}(Y) \subset \mathsf{AN}(Y) \subset \mathsf{ND}(Y)$ and $\mathsf{CH}(Y) \subset \mathsf{DE}(Y)$. There is an additional node $E$ in the graph to denote the different environments (or contexts) (Mooij et al., 2020). A set of variables/predictors $\mathbf{S}$ is invariant if $Y \perp\!\!\!\perp E | \mathbf{S}$. As is common in the causal inference literature, we assume (1) no hidden confounder, (2) no intervention on $Y$, (3) no feedback between variables, (4) independence of mechanisms, and (5) independent additive noise structural causal models (SCMs), with no noise-free mechanisms and duplicate variables. We further assume that (6) the additive noise variables in the SCM have unbounded support (e.g. Gaussian noise). Like prior work (Mogensen et al., 2022; Mooij et al., 2020), we assume that (7) $E$ is exogenous (i.e., $E$ has no parent).

### 3.2 ICP and IAS

ICP exploits distribution shifts between different environments $E$ to learn a subset of $\mathsf{PA}(Y)$ (Peters et al., 2016). Given a test for the hypothesis $H_{0,\mathbf{S}}$ that $\mathbf{S}$ is invariant, ICP outputs the intersection of all invariant subsets. Specifically, $\hat{\mathbf{S}}_{\mathrm{ICP}} := \bigcap_{\mathbf{S}:H_{0,\mathbf{S}} \text{ not rejected}} \mathbf{S}$. In contrast, IAS tries to identify a subset of $\mathsf{AN}(Y)$ (Mogensen et al., 2022). Let $\widehat{\mathcal{I}}$ be the set of all sets $\mathbf{S}$ for which $H_{0,\mathbf{S}}$ is not rejected. Define $\widehat{\mathcal{MI}} := \left\{ \mathbf{S} \in \widehat{\mathcal{I}} \mid \forall \mathbf{S}' \subsetneq \mathbf{S} : \mathbf{S}' \notin \widehat{\mathcal{I}} \right\}$. Thus, $\hat{\mathbf{S}}_{\mathrm{IAS}} := \bigcup_{\mathbf{S} \in \widehat{\mathcal{MI}}} \mathbf{S}$. Although Peters et al. (2016) proposed two different tests for invariance, $H_{0,\mathbf{S}}$, the approximate test based on residuals of a predictor (Method II) is usually used because of its speed. Specifically, this statistical test checks whether the distribution (means and variances) of the predictor's errors across environments $E$ are the same. The predictor is fitted using the data from all environments (see Appendix A for more details).

### 3.3 Finding causal parents by minimizing error

Instead of taking the intersection or union of invariant subsets, we rely on the concept of minimum mean squared error predictor to find the causal parents. The mean squared error (MSE) of a predictior $\hat{Y}(\mathbf{X})$ with input variables $\mathbf{X}$ of target $Y$ is $\mathbb{E}_{\mathbf{X}Y}(Y - \hat{Y}(\mathbf{X}))^2$, where $\mathbb{E}$ denotes expectation. Let the minimum MSE achieved by an optimal predictor $\hat{Y}_M(\mathbf{X})$ with input variables $\mathbf{X}$ be $\mathsf{MMSE}(\mathbf{X})$. Note that this is always with respect to predicting a target variable $Y$. It is well-known that, if the minimization is unconstrained, the $\mathsf{MMSE}(\mathbf{X})$ is the conditional expectation $\mathbb{E}_{Y|\mathbf{X}}(Y|X)$, and the corresponding MMSE is equal to $\mathbb{E}_{\mathbf{X}}\mathrm{Var}_{Y|\mathbf{X}}(Y|X)$ (Leon-Garcia, 1994), where Var denotes variance. In practice, we will assume that a

large-capacity model, trained on enough data with least square loss, can approximate this MMSE estimator. The following provides the theoretical grounding for MMSE-ICP.

### 3.3.1 Minimum mean squared error inequality

**Lemma 3.1** (Error Inequality). *Let $\mathbf{X}_1$ and $\mathbf{X}_2$ denote two sets of variables, not necessarily mutually exclusive. Then:* $\mathsf{MMSE}(\mathbf{X}_1 \cup \mathbf{X}_2) \leq \mathsf{MMSE}(\mathbf{X}_1)$. *Equality holds if* $Y \perp\!\!\!\perp \mathbf{X}_2 | \mathbf{X}_1$.

*Proof.* $\mathsf{MMSE}(\mathbf{X}_1 \cup \mathbf{X}_2) = \mathbb{E}_{\mathbf{X}_1, \mathbf{X}_2} \left[ \min_{\hat{Y}} \mathbb{E}_{Y | \mathbf{X}_1, \mathbf{X}_2} (Y - \hat{Y})^2 \right] = \mathbb{E}_{\mathbf{X}_1} \mathbb{E}_{\mathbf{X}_2 | \mathbf{X}_1} \left[ \min_{\hat{Y}} \mathbb{E}_{Y | \mathbf{X}_1, \mathbf{X}_2} (Y - \hat{Y})^2 \right]$.

Let $\hat{Y}_M(\mathbf{X}_1, \mathbf{X}_2) := \arg\min_{\hat{Y}} \mathbb{E}_{Y | \mathbf{X}_1, \mathbf{X}_2} (Y - \hat{Y})^2$, then:

$$\mathbb{E}_{Y | \mathbf{X}_1, \mathbf{X}_2} (Y - \hat{Y}_M(\mathbf{X}_1, \mathbf{X}_2))^2 \leq \mathbb{E}_{Y | \mathbf{X}_1, \mathbf{X}_2} (Y - \hat{Y})^2, \ \ \forall \hat{Y} \tag{1}$$

Since expectation is monotonic,

$$\Rightarrow \mathbb{E}_{\mathbf{X}_2 | \mathbf{X}_1} \left[ \mathbb{E}_{Y | \mathbf{X}_1, \mathbf{X}_2} (Y - \hat{Y}_M(\mathbf{X}_1, \mathbf{X}_2))^2 \right] \leq \mathbb{E}_{\mathbf{X}_2 | \mathbf{X}_1} \left[ \mathbb{E}_{Y | \mathbf{X}_1, \mathbf{X}_2} (Y - \hat{Y})^2 \right] = \mathbb{E}_{Y | \mathbf{X}_1} (Y - \hat{Y})^2, \ \ \forall \hat{Y} \tag{2}$$

Since Equation 2 holds for every $\hat{Y}$, it is also true for $\hat{Y}_M(\mathbf{X}_1) = \arg\min_{\hat{Y}} \mathbb{E}_{Y | \mathbf{X}_1} (Y - \hat{Y})^2$,

$$\Rightarrow \mathbb{E}_{\mathbf{X}_1} \mathbb{E}_{\mathbf{X}_2 | \mathbf{X}_1} \left[ \mathbb{E}_{Y | \mathbf{X}_1, \mathbf{X}_2} (Y - \hat{Y}_M(\mathbf{X}_1, \mathbf{X}_2))^2 \right] \leq \mathbb{E}_{\mathbf{X}_1} \left[ \mathbb{E}_{Y | \mathbf{X}_1} (Y - \hat{Y}_M(\mathbf{X}_1))^2 \right] = \mathsf{MMSE}(\mathbf{X}_1) \tag{3}$$

$$\Rightarrow \mathsf{MMSE}(\mathbf{X}_1 \cup \mathbf{X}_2) \leq \mathsf{MMSE}(\mathbf{X}_1) \tag{4}$$

Equality occurs when $\hat{Y}_M(\mathbf{X}_1, \mathbf{X}_2) = \hat{Y}_M(\mathbf{X}_1)$, i.e.,

$$\arg\min_{\hat{Y}} \mathbb{E}_{Y | \mathbf{X}_1, \mathbf{X}_2} (Y - \hat{Y})^2 = \arg\min_{\hat{Y}} \mathbb{E}_{Y | \mathbf{X}_1} (Y - \hat{Y})^2, \ \ \forall \mathbf{X}_1, \mathbf{X}_2 \tag{5}$$

If $Y \perp\!\!\!\perp \mathbf{X}_2 | \mathbf{X}_1$ then one can easily show that Equation 5 holds. $\qquad \square$

**Corollary 3.2.** *In the causal DAG that satisfies the assumptions stated in Section 3.1, for all subset $\mathbf{S}$ of* $\mathsf{ND}(Y)$*:*

- $\mathsf{MMSE}(\mathbf{S}) \geq \mathsf{MMSE}(\mathsf{PA}(Y))$,

- $\mathsf{MMSE}(\mathbf{S}) = \mathsf{MMSE}(\mathsf{PA}(Y))$, *if and only if* $\mathsf{PA}(Y) \subset \mathbf{S}$, *and*

- $\mathsf{MMSE}(\mathbf{S}) > \mathsf{MMSE}(\mathsf{PA}(Y))$, *if and only if* $\mathsf{PA}(Y) \not\subset \mathbf{S}$.

*Proof.* The first inequality is an immediate corollary of Lemma 3.1. Since $\mathbf{S} \subset \mathsf{ND}(Y)$, it follows from the Causal Markov condition that $Y \perp\!\!\!\perp \mathbf{S} | \mathsf{PA}(Y)$. By Lemma 3.1, $\mathsf{MMSE}(\mathbf{S} \cup \mathsf{PA}(Y)) \leq \mathsf{MMSE}(\mathbf{S})$. Furthermore, also by Lemma 3.1, $\mathsf{MMSE}(\mathsf{PA}(Y) \cup \mathbf{S}) = \mathsf{MMSE}(\mathsf{PA}(Y))$. Hence, $\mathsf{MMSE}(\mathbf{S}) \geq \mathsf{MMSE}(\mathsf{PA}(Y))$.

It is easy to show that, in the assumed DAG, $\mathsf{MMSE}(\mathsf{PA}(Y)) = \sigma_Y^2$, where $\sigma_Y^2$ is the variance of the independent additive noise for $Y$. Furthermore, for any $\mathbf{S}$ which is a subset of $\mathsf{ND}(Y)$, $\mathsf{MMSE}(\mathbf{S}) = \mathbb{E}_{\mathbf{S}} \mathrm{Var}_{Y | \mathbf{S}}(f_Y(\mathsf{PA}(Y)) | \mathbf{S}) + \sigma_Y^2$, where $f_Y(\mathsf{PA}(Y))$ is the functional mapping from $Y$'s parents to $Y$ in the SCM. It can be shown that in the assumed noisy SCM where the noise distribution has unbounded support (e.g. Gaussian noise), $\mathrm{Var}_{Y | \mathbf{S}}(f_Y(\mathsf{PA}(Y)) | \mathbf{S}) > 0$ if and only if $\mathsf{PA}(Y) \not\subset \mathbf{S}$ and $\mathrm{Var}_{Y | \mathbf{S}}(f_Y(\mathsf{PA}(Y)) | \mathbf{S}) = 0$ if and only if $\mathsf{PA}(Y) \subset \mathbf{S}$. $\qquad \square$

For example, in Figure 1, $\mathsf{MMSE}(\{X_2\}) > \mathsf{MMSE}(\{X_1, X_2\}) = \mathsf{MMSE}(\{X_1\})$. Furthermore, since $\{X_1, X_2\}$ has 2 elements while $\{X_1\}$ has only 1 element, we can deduce that $\{X_1\}$ is the causal parent and not $\{X_2\}$.

---

**Algorithm 1:** MMSE-ICP

---

**Input:** $E, X_1, X_2, \ldots, X_d, Y$
**Output:** Potential parents of $Y$
**1** **if** *is_invariant*$(E, \mathbf{X}, Y, \emptyset)$ **then** **return** $\emptyset$
**2** candidates $= []$
**3** **for** $\mathbf{S} \subset \{X_1, X_2, \ldots, X_d\}$ *in increasing cardinality order* **do**
**4**     **if** $\exists \mathbf{S}' \in$ candidates *such that* $\mathbf{S}' \subset \mathbf{S}$ **then** **continue**
**5**     **if** *is_invariant*$(E, \mathbf{X}, Y, \mathbf{S})$ **then** candidates = candidates $+ [\mathbf{S}]$
**6** **return** $\arg\min_{\mathbf{S} \in \text{candidates}} \text{MMSE}(\mathbf{S})$

---

### 3.3.2 Combining MMSE and invariance test

Algorithm 1 combines the error inequality with the invariance test to find causal parents. Theorem 3.4 shows that Algorithm 1 can identify all causal parents of $Y$ as long as $\mathsf{PA}(Y) \subset \mathsf{DE}(E)$. This implies that Algorithm 1 usually requires fewer interventions than the number of variables for complete identification. For example, a single intervention at an upstream variable that affects all causal parents of $Y$ is sufficient. This is much more favorable than ICP, which may need as many interventions at as the number of variables (Peters et al., 2016).

**Lemma 3.3.** *If* $\mathbf{S}_1$ *is invariant, then* $\mathbf{S}_2 = \mathbf{S}_1 \cap (\mathsf{DE}(E) \cap \mathsf{ND}(Y))$ *is also invariant. In other words, removing all nodes in* $\mathbf{S}_1$ *that are not in* $\mathsf{DE}(E) \cap \mathsf{ND}(Y)$ *from* $\mathbf{S}_1$ *results in another invariant set.*

*Proof.* Since $\mathbf{S}_1$ is invariant, $Y \perp\!\!\!\perp E | \mathbf{S}_1$ so $\mathbf{S}_1$ blocks all paths between $E$ and $Y$ by d-separation (Pearl, 2009). If a node $X$ is in $\mathbf{S}_1$ but not in $\mathsf{DE}(E) \cap \mathsf{ND}(Y)$, then one of these two cases must hold.

1. $X$ is not on any blocking path so $\mathbf{S}_1 \setminus \{X\}$ still blocks all paths between $E$ and $Y$.

2. $X$ is on a blocking path between $E$ and $Y$. Since $E$ has no parent (see Section 3.1), $X \in \mathsf{DE}(E) \cap \mathsf{DE}(Y)$. Thus, this blocking path is out-going from $Y$. Removing all nodes on this path will keep the path blocked. The removed nodes are descendants of $Y$ so they are not in $\mathsf{DE}(E) \cap \mathsf{ND}(Y)$.

In both cases, invariance is not affected by excluding $X$. Thus, excluding nodes in $\mathbf{S}_1$ that are not in $\mathsf{DE}(E) \cap \mathsf{ND}(Y)$ results in $\mathbf{S}_2$ that is also invariant. $\square$

**Theorem 3.4** (Identifiability). *Given that all tests of invariance return the correct results and* $\mathsf{PA}(Y) \subset \mathsf{DE}(E)$, *Algorithm 1 will always find the set of causal parents* $\mathsf{PA}(Y)$.

Theorem 3.4 follows from Corollary 3.2 and Lemma 3.3. The loop in Algorithm 1 finds invariant subsets of $\mathsf{DE}(E) \cap \mathsf{ND}(Y)$ which include $\mathsf{PA}(Y)$ when $\mathsf{PA}(Y) \subset \mathsf{DE}(E)$. Amongst these subsets, $\mathsf{PA}(Y)$ has minimum MSE and is the most compact. The solution is unique (no other subset has the same MSE and cardinality) because other subsets with minimum MSE are all strict supersets of $\mathsf{PA}(Y)$ so they have larger cardinality than $\mathsf{PA}(Y)$.

Lemma 3.3 implies that for any invariant set $\mathbf{S}_1$ that is not a subset of $\mathsf{DE}(E) \cap \mathsf{ND}(Y)$, there exists an invariant set $\mathbf{S}_2$ that is a subset of $\mathbf{S}_1$ (having smaller cardinality than $\mathbf{S}_1$'s) and of $\mathsf{DE}(E) \cap \mathsf{ND}(Y)$. Since Algorithm 1 iterates through the sets of variables in increasing cardinality order, it will first find $\mathbf{S}_2$ and add $\mathbf{S}_2$ to the list. It will later exclude $\mathbf{S}_1$ because $\mathbf{S}_2$ is a subset of $\mathbf{S}_1$. Thus, all invariant sets that are not subsets of $\mathsf{DE}(E) \cap \mathsf{ND}(Y)$ will be excluded. $\mathsf{PA}(Y)$ will always be in the list because no subset of $\mathsf{PA}(Y)$ is invariant.

### 3.4 Faster search sequence

Although Algorithm 1 should be more accurate than ICP, it still needs to run an exponential number of invariance tests. We propose Algorithm 2, which is faster and consists of two stages.

---

**Algorithm 2:** fastICP

**Input:** $E, X_1, X_2, \ldots, X_d, Y, \mathsf{MaxDepth}$
**Output:** Potential parents of $Y$

**1 if** *is_invariant*$(E, \mathbf{X}, Y, \emptyset)$ **then return** $\emptyset$
   // Find invariant set
**2** $\mathbf{S} = \{X_1, \ldots, X_d\}$
**3 while** *not is_invariant*$(E, \mathbf{X}, Y, \mathbf{S})$ **do**
**4**     candidates $= \{\mathbf{S}' \subset \mathbf{S} : |\mathbf{S}'| \geq |\mathbf{S}| - \mathsf{MaxDepth}\}$
**5**     **if** *no* candidates **then return** $\emptyset$
**6**     $\mathbf{S} = \arg\min_{\mathbf{S}' \in \text{candidates}} \mathsf{statDependency}(Y, \mathbf{S}')$

   // Prune invariant set
**7 while** *True* **do**
**8**     candidates $= []$
**9**     **for** $Z \in \mathbf{S}$ **do**
**10**       **if** *is_invariant*$(E, \mathbf{X}, Y, \mathbf{S} \setminus \{Z\})$ **then**
**11**         candidates $=$ candidates $+ [\mathbf{S} \setminus \{Z\}]$

**12**     **if** *no* candidates **then break**
**13**     **else** $\mathbf{S} = \arg\min_{\mathbf{S}' \in \text{candidates}} \mathsf{MMSE}(\mathbf{S}')$

**14 return** $\mathbf{S}$

---

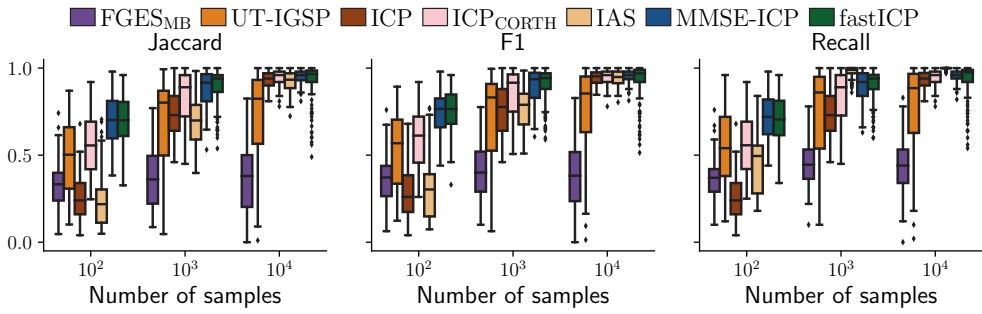

Figure 2: Performance when $N_{\text{int}} = d = 6$ (Table 1, No. 1). Linear simulation. Reference set: $\mathsf{PA}(Y)$.

Stage 1 aims to find the largest invariant set $\mathbf{S}$ that includes $\mathsf{PA}(Y)$. Starting with the set of all variables, it removes potential colliders and their descendants, which might be unblocking paths from $E$ to $Y$, using the heuristic presented in Cheng et al. (1998). The heuristic assumes that blocking a path by removing some nodes from the conditioning set decreases the statistical dependency between $E$ and $Y$ (Cheng et al., 1998). In Algorithm 2, $\mathsf{MaxDepth}$ is a hyper-parameter for the maximum number of nodes to be removed at one time. The statistical dependency can be measured using the invariance test of Peters et al. (2016) where the lower the statistical dependency, the higher the probability of being invariant. Stage 2, based on Proposition 3.5, removes variables from $\mathbf{S}$ one-by-one while maintaining invariance. The algorithm terminates when $\mathbf{S} = \mathsf{PA}(Y)$.

**Proposition 3.5.** *We are given that* $\mathsf{PA}(Y) \subset \mathsf{DE}(E)$, $\mathbf{S}$ *is invariant and has the lowest MMSE amongst subsets with the same cardinality, and* $\mathsf{PA}(Y) \subset \mathbf{S}$. *For each* $Z \in \mathbf{S}$, *let* $\mathbf{S}_{-Z} := \mathbf{S} \setminus \{Z\}$. *If no subset* $\mathbf{S}_{-Z}$ *of $S$ is invariant, then* $\mathbf{S} = \mathsf{PA}(Y)$. *Otherwise,* $\mathsf{PA}(Y) \subset \arg\min_{\text{invariant } \mathbf{S}_{-Z}} \mathsf{MMSE}(\mathbf{S}_{-Z})$.

*Proof.* If no subset $\mathbf{S}_{-Z}$ of $S$ is invariant, all nodes of $\mathbf{S}$ must be on blocking paths because removing any node $Z$ from $\mathbf{S}$ leads to loss of invariance (unblocks a path from $E$ to $Y$). If multiple nodes are on the same blocking path, one node (e.g. $W$) can be removed and $\mathbf{S}_{-W}$ is still invariant which is contradictory. Thus, $Z \in \mathsf{PA}(Y)$ because otherwise, exchanging $Z$ with the parent in the same blocking path would result in a

Table 1: Different setups with different number of nodes $(X_1, \ldots, X_d, Y, E)$, graph densities, number of interventions $(N_{\text{int}})$, and type of interventions. MB: Markov Blanket

| No. | $d$ | Density | MB size | $N_{\text{int}}$ | Type |
|-----|-----|---------|---------|------------------|------|
| 1. | 6 | 0.240 | 2.98 | 6 | Perfect |
| 2. | 6 | 0.145 | 2.58 | 1,2,3 | Perfect |
| 3. | 6 | 0.158 | 2.89 | 1 | Imperfect |
| 4. | 6 | 0.153 | 2.86 | 1 | Noise |
| 5. | 100 | 0.010 | 3.30 | 1–5 | Perfect |
| 6. | 100 | 0.050 | 26.99 | 1–5 | Perfect |

subset with the same cardinality but with smaller MMSE, which is contradictory. Since $Z \in \mathsf{PA}(Y), \forall Z \in \mathbf{S}$, then $\mathbf{S} = \mathsf{PA}(Y)$.

Suppose there exists a $Z \in \mathbf{S}$ where $\mathbf{S}_{-Z}$ is invariant. Let's assume that $\mathsf{PA}(Y) \nsubseteq \mathbf{S}_{-Z'} := \arg\min_{\text{invariant } \mathbf{S}_{-Z}} \mathsf{MMSE}(\mathbf{S}_{-Z})$.
Since $\mathsf{PA}(Y) \nsubseteq \mathbf{S}_{-Z'}$ but $\mathsf{PA}(Y) \subset \mathbf{S}$ so $Z' \in \mathsf{PA}(Y)$. Besides, $\mathbf{S}_{-Z'}$ is invariant but $Z' \notin \mathbf{S}_{-Z'}$, so there must be another node $T \in \mathbf{S}_{-Z'}$ that is on the same blocking path as $Z'$.
Let $\mathbf{S}' := \mathbf{S} \setminus \{Z', T\}$. Consequently, $\mathbf{S}_{-Z'} = \mathbf{S} \setminus \{Z'\} = \mathbf{S}' \cup \{T\}$ and $\mathbf{S}_{-T} = \mathbf{S} \setminus \{T\} = \mathbf{S}' \cup \{Z'\}$.
Since $T$ and $Z'$ are on the same blocking path, $\mathbf{S}_{-T}$ is also invariant. In addition, because $Z' \in \mathsf{PA}(Y)$, $T$ must precede $Z'$, i.e. $Y \perp\!\!\!\perp T | \mathbf{S}' \cup \{Z'\} \Leftrightarrow Y \perp\!\!\!\perp T | \mathbf{S}_{-T}$. As a result, by Lemma 3.1 and Corollary 3.2,

$$\mathsf{MMSE}(\mathbf{S}_{-T}) = \mathsf{MMSE}(\mathbf{S}_{-T} \cup \{T\}) = \mathsf{MMSE}(\mathbf{S}) = \mathsf{MMSE}(\mathbf{S}_{-Z'} \cup \{Z'\}) < \mathsf{MMSE}(\mathbf{S}_{-Z'}).$$

This is contradictory because $\mathsf{MMSE}(\mathbf{S}_{-Z'}) < \mathsf{MMSE}(\mathbf{S}_{-T})$ by definition. Hence, $\mathsf{PA}(Y) \subset \mathbf{S}_{-Z'}$. □

### 3.5 Complexity Analysis

For a problem with $V$ variables, the complexity of ICP and MMSE-ICP is $O(2^V)$ as the number of invariance tests run is exponential. Stage 1 of fastICP is $O(V * 2^{\mathsf{MaxDepth}})$ because of the number of candidates checked grows exponentially with MaxDepth. Stage 2 is $O(V^2)$. Thus, the complexity of fastICP is $O(V * 2^{\mathsf{MaxDepth}} + V^2)$. When MaxDepth is less than $V$, fastICP is faster than ICP.

## 4 Simulation Experiments

### 4.1 Data

We generate synthetic datasets according to the assumptions in Section 3.1 (additive noise model with i.i.d. Gaussian noise). Similar to Peters et al. (2016); Mogensen et al. (2022), we consider 2-environment setups (observational and interventional). In each setup, 100 graphs are randomly generated. In a graph, beside the target node $Y$ and the environment indicator node $E$, there are $d$ additional nodes $\{X_1, \ldots, X_d\}$. The edges from $E$ to a subset of $\{X_1, \ldots, X_d\}$ specify the nodes that may be intervened on. The number of interventions $(N_{\text{int}})$ varies between different setups. We do not enforce the constraint that $\mathsf{PA}(Y) \subset \mathsf{DE}(E)$ to test the robustness of the proposed methods when this assumption is invalid. Values of a node is generated based on its parents' values, through linear or nonlinear functions. Beside linear simulations whereby all functions are linear, we also include nonlinear simulations (see Appendix B). For each graph, 50 sets of coefficients of the generative functions are drawn randomly. For each set of coefficients, we sample 4 datasets with different sample sizes $(10^2, 10^3, 10^4, 10^5)$. The data are standardized along the causal order to prevent shortcut learning (Reisach et al., 2021). In the interventional environment, interventions are applied to a random subset of children of $E$. We consider 3 types of interventions: (1) perfect intervention which severs all causal dependencies from parents, (2) imperfect intervention which modifies causal relations between a node and its parents, and (3) noise intervention in which the intervened variable's noise variance changes (Cooper & Yoo, 1999; Peters et al., 2016). The different setups are summarized in Table 1.

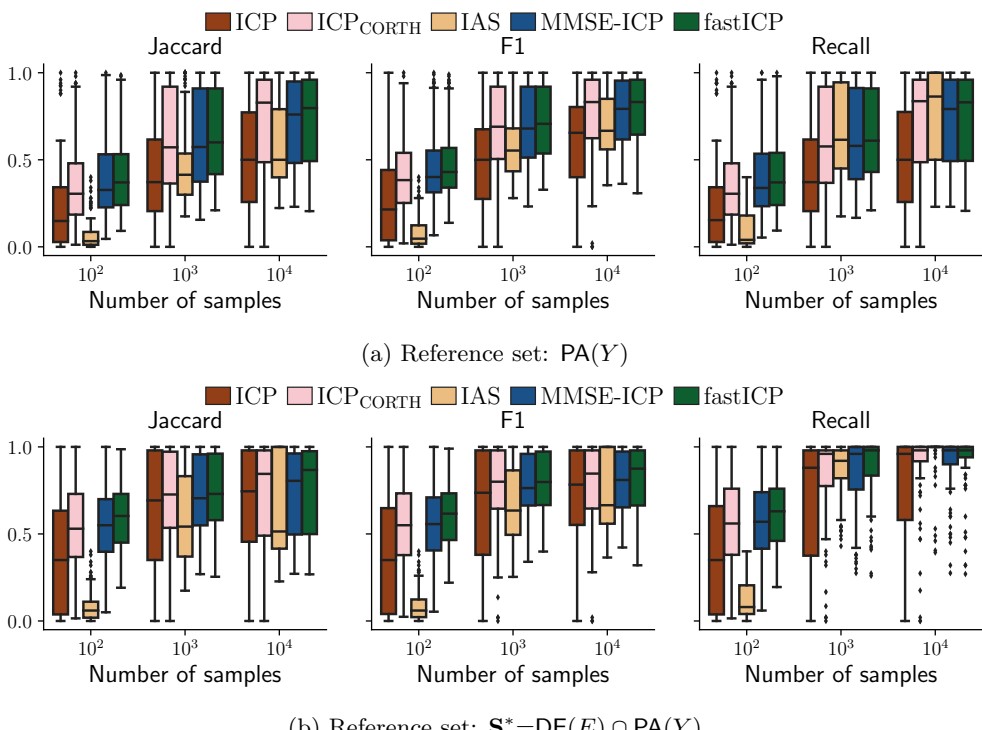

(a) Reference set: $\mathsf{PA}(Y)$

(b) Reference set: $\mathbf{S}^*=\mathsf{DE}(E)\cap\mathsf{PA}(Y)$

Figure 3: Performance when $N_{\text{int}}=1; d=6$ (Table 1, No. 2). Linear simulation. See Appendix C.1 for results when $N_{\text{int}}=2$ or $N_{\text{int}}=3$.

## 4.2 Baselines and implementation details

We compare MMSE-ICP and fastICP against ICP (Peters et al., 2016),IAS (Mogensen et al., 2022),fGES-MB (Ramsey et al., 2017),UT-IGSP (Squires et al., 2020),and ICP-CORTH. ICP-CORTH uses ICP to remove false positives in CORTH's output (Soleymani et al., 2022). Since CORTH assumes $Y$ is a linear combination of its parents (similar to the simulation setup), ICP-CORTH is a competitive baseline. To test for invariance, MMSE-ICP and fastICP use the same test as ICP and IAS. Specifically, the test checks whether the mean and variance of the prediction residuals (of a regression model) is equal across environments. Linear regression is used for linear simulations while gradient boosted tree is used for nonlinear simulations (see Appendix A). To keep the runtime of the invariance test comparable to ICP and IAS, MMSE is estimated by averaging the prediction residuals (see Appendix C.2 for experiments to test the robustness of MMSE estimation).

The MaxDepth hyper-parameter is set at 2. For ICP, IAS, UT-IGSP, MMSE-ICP, and fastICP, the significance level $\alpha$ is set at 0.05. The parameters of IAS ($C$ and $\alpha_0$) are set according to the original paper (Mogensen et al., 2022). Since it is intractable to search exhaustively using ICP, IAS and MMSE-ICP for large $d$, their search scopes are restricted in these cases. In particular, ICP and MMSE-ICP only search within an estimated MB of size 10. The 10 variables that are considered for further analysis are determined via L2-boosting (Friedman, 2001; Bühlmann & Yu, 2003; Hothorn et al., 2010). IAS only tests for sets with a single element.

*Jaccard similarity* and *F1-score* are used as metrics. The prediction is compared against a reference set, which could be either (1) the full set parents $\mathsf{PA}(Y)$ and (2) the set of perturbed parents $\mathbf{S}^*=\mathsf{DE}(E)\cap\mathsf{PA}(Y)$. Since IAS finds ancestors instead of parents, we also look at the parents' *Recall* rate.

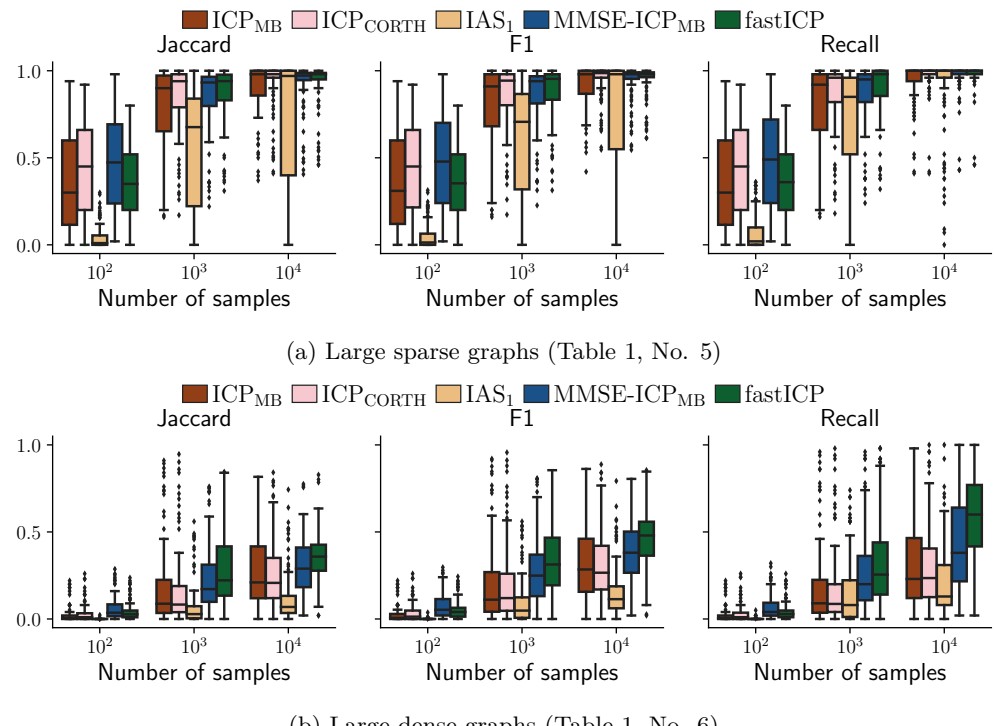

(a) Large sparse graphs (Table 1, No. 5)

(b) Large dense graphs (Table 1, No. 6)

Figure 4: Performance for large graphs. Reference set: $\mathbf{S}^*$. Also see Figure 11 and 12 in the Appendix.

## 4.3 Results from linear simulations

When, $N_{\text{int}} = d = 6$, invariance-based algorithms should be able to discover all parents. Figure 2 shows their results in this setting. With sufficient samples, invariance-based algorithms outperform fGES-MB (observational constraint-based CD) and UT-IGSP. When the number of interventions is limited (i.e. $N_{\text{int}} < d$, Table 1, No. 2–6), identifying all direct parents using invariance is more challenging since some invariance subsets may be strict subset of the parents, i.e. $|\mathbf{S}^*| < |\mathsf{PA}(Y)|$. Due to the strict inequality, we only benchmark our approaches against invariance-based algorithms. For perfect interventions, MMSE-ICP and fastICP achieve similar performance and outperform the baselines in both Jaccard similarity and F1-score (Figure 3). The recall of our approaches is the same as IAS's and higher than ICP's. Figure 3 shows

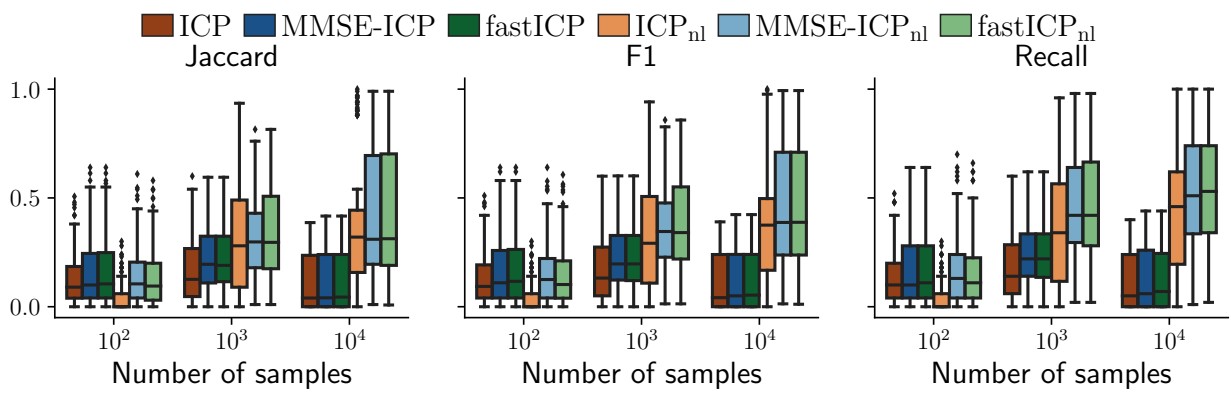

Figure 5: Performance for nonlinear simulation. $N_{\text{int}}=1; d=6$. Reference set: $\mathbf{S}^*$. *nl*: using nonlinear regression in invariance test. Also see Figure 18 in the Appendix.

that our approaches often found most of the perturbed parents ($\mathbf{S}^*$). The same trends are observed for imperfect interventions (Figure 16), noise interventions (Figure 17), and large sparse graphs (Figure 4a). They hold for large sparse graphs since the estimated MB of size 10 is sufficient to cover the true MB (see Table 1). Although MMSE-ICP and fastICP should perform similarly when the former can exhaustively test all subsets, in practice, their performance may differ as they test the subsets in different orders (increasing cardinality order for MMSE-ICP and decreasing cardinality order for fastICP). With imperfect invariance test, the order of testing may impact the performance since some tests may more likely fail than others.

For large dense graph (Figure 4b), MMSE-ICP and fastICP still outperform the baselines. Since the estimated MB is much smaller than the true MB, methods that cannot search exhaustively will miss out many parents. ICP-CORTH is worse than ICP probably because CORTH's assumption that the target Y has no children is more likely to be wrong. When the graph is large and dense, MMSE-ICP cannot search exhaustively so the ability to search through all nodes gives fastICP an edge over MMSE-ICP.

### 4.4 Computation time analysis

To analyze the runtime complexity of invariance-based algorithms, we recorded the time each algorithm took when the number of nodes ($d$) varies. The data were generated from linear SCMs as outlined in Section 4.1, but with varying $d$ and $N$ (number of samples) fixed at 1000. Table 2 reports the numbers of seconds elapsed when executing on an AMD EPYC 7642 CPU core (@ 2.3GHz). For this benchmark, since the official ICP implementation was in R, we employed a re-implementation in Python for a fair comparison. The algorithms search through the full set of covariates (i.e. no pre-selection of variables using Markov Blanket estimation).

Table 2: The average runtime in seconds of different algorithms for varying $d$.

| Method | Language | $d = 6$ | $d = 9$ | $d = 12$ | $d = 15$ | $d = 18$ | $d = 21$ |
|---|---|---|---|---|---|---|---|
| ICP | R | $0.16_{\pm 0.01}$ | $1.78_{\pm 0.06}$ | $17.66_{\pm 0.69}$ | $339.5_{\pm 53.2}$ | $> 3600$ | $> 3600$ |
| ICP | Python | $0.06_{\pm 0.01}$ | $0.41_{\pm 0.08}$ | $3.28_{\pm 1.06}$ | $30.2_{\pm 10.8}$ | $224.6_{\pm 67.2}$ | $1794_{\pm 593.}$ |
| MMSE-ICP | Python | $0.06_{\pm 0.01}$ | $0.42_{\pm 0.08}$ | $3.30_{\pm 1.06}$ | $29.9_{\pm 10.6}$ | $225.5_{\pm 66.1}$ | $1806_{\pm 618.}$ |
| fastICP | Python | $0.03_{\pm 0.00}$ | $0.07_{\pm 0.00}$ | $0.12_{\pm 0.00}$ | $0.20_{\pm 0.01}$ | $0.28_{\pm 0.01}$ | $0.39_{\pm 0.02}$ |

From these runtime values, we confirm that ICP and MMSE-ICP both have exponential complexity in $d$. Furthermore, the additional computation overhead of MMSE-ICP as compared to ICP is negligible. In contrast, the runtime of fastICP is polynomial (quadratic) in $d$ so fastICP is much more scalable than ICP and MMSE-ICP.

### 4.5 Results from nonlinear simulations

For nonlinear simulation, our approaches are still better than ICP (Figure 5 and Appendix C.4). As predicted by Heinze-Deml et al. (2018), it is sometime possible to use the linear invariance test (invariance test with linear regression) to find causal parents in nonlinear simulations. However, it is preferable to use a nonlinear invariance test to obtain more accurate results if runtime is not a constraint.

## 5 Experiment on gene expression data

### 5.1 Data

We apply our methods to a real-world large-scale yeast gene expression dataset with 6170 variables (Kemmeren et al., 2014). There are 160 observational samples and 1479 interventional samples. Each interventional sample corresponds to an experiment where single $X_k$ has been perturbed. Following Peters et al. (2016); Mogensen et al. (2022), we assume that a direct causal effect $X \rightarrow Y$ exists (true positive) if the

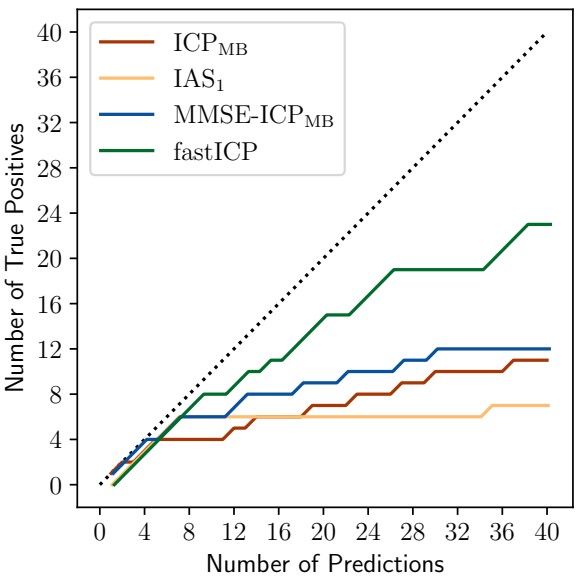

Figure 6: Performance on genes' parents prediction. The dotted line delineates perfect accuracy.

expression level of gene $Y$ after intervening on gene $X$ lies in the 1% lower or upper tail of the observational distribution of gene $Y$.

Since neither the ground-truth graph nor a separate set of validation data is available, we must use the same data for validation. Hence, we employ the same cross-validation scheme used in the original ICP paper to prevent information leakage (Peters et al., 2016). Specifically, when predicting whether $X_k$ is a parent of $Y$, we do not include the sample when intervening on $X_k$ (if the sample exists) in the data used for inference. The interventional samples are split into 3 folds. In each fold, two thirds of the interventional samples not containing $X_k$ are used as interventional data, and remaining interventional data are used for validation. Thus, for each target, we need to run the algorithms the same number of times s the number of folds. Additionally, when looking for potential causes of $Y$, we exclude samples corresponding to intervention on $Y$ (if it exists).

## 5.2 Baselines and implementation details

ICP and IAS are used as baselines as they give confidence estimates for individual predicted parents/ancestors. The same linear invariance test is used although the threshold is set at $\alpha = 0.01$, following Peters et al. (2016). L2-boosting (Friedman, 2001; Bühlmann & Yu, 2003; Hothorn et al., 2010) is used to estimate the MB of size 10 for ICP and MMSE-ICP. Even though fastICP can search exhaustively in simulations of 100 variables, it would be too slow for this problem. Hence, we restrict the fastICP search scope to an estimated MB of 100 variables. ICP-CORTH is excluded because it takes more than 3 days for one gene so obtaining the result for 6170 genes would take more than 18000 days. Instead scoring the methods' output at fixed thresholds, we ranked the set of predicted causal relations and score the most confident predicted relations.

## 5.3 Results

Figure 6 shows how the number of true positives vary as the methods are allowed to make more predictions. When the number of allowed predictions is lower than 8, the performance of the methods are very similar. However, when the number of predictions is more than 10, MMSE-ICP is generally more accurate than ICP. Moreover, fastICP is the best approach for this task as it can afford to search more widely for the parents of

each gene. fastICP obtained 20 true positives when making 30 predictions (66% success probability), setting a new state-of-the-art (SOTA) for this benchmark.

## 6 Discussion

While using only observational data for causal discovery (CD) is common, interventional data with mechanism changes are very valuable for inferring causal relations. In fact, changes (e.g. different equipments, locations, demographics, weathers) often arise naturally in distribution-shift data. Given their abundance, harnessing changes for CD would reveal insights useful for understanding and manipulating complex systems. However, exploiting changes in general is difficult as they are often imperfect interventions (merely altering the mechanism generating a variable) affecting some unknown variables. In contrast, controlled experiments are perfect interventions that fix designated variables to predetermined values (Tian & Pearl, 2001; Eberhardt & Scheines, 2007). As such, invariance-based CD methods are appealing because they can work with unknown interventions.

Like other invariance-based CD methods, ours also make no assumption about where interventions are and what precisely the effect of interventions may be. We only assume that there is no intervention on the target. This make our approaches appealing in many problems whereby specifying what an intervention or change of environment actually means is difficult. In addition, unlike ICP which needs a sufficient number of interventions to identify all parents (Rosenfeld et al., 2021; Mogensen et al., 2022), our approaches can identify all parents even if they are only indirectly effected by interventions. Thus, our methods can be applied to problems where it is logistically challenging to exhaustively intervene on many variables (Uhler, 2024). Besides, our work may have implications for robust representation learning. Building ML models with invariant representations has become popular recently since they can generalize better to distribution-shift data (Arjovsky et al., 2019; Rosenfeld et al., 2021; Nguyen et al., 2024c;b). It may be possible to build even more accurate invariant representations by identifying causal variables (Subbaswamy et al., 2019). Thus, as an efficient way to find causal variables from distribution-shift data, fastICP can be a key to more adaptive ML models.

In this work, we proposed two algorithms: MMSE-ICP which has similar runtime but better recall than ICP; and fastICP which is more scalable. Both approaches outperform multiple baselines in simulations. fastICP also achieves SOTA result on the large-scale Kemmeren et al. (2014)'s benchmark. MMSE-ICP and fastICP are based on general theoretical results are orthogonal to the implementations of invariance test and MSE estimation. There are several unaddressed questions that are left as future work.

First, despite its speed, fastICP still has worst-case exponential complexity that is controlled by the MaxDepth parameter. The larger the MaxDepth, the closer fastICP is to ICP (i.e. testing exponential number of subsets). Hence, setting a smaller MaxDepth will result in greater speed-up, with an increased risk for inaccurate results. In our experiments, MaxDepth of 2 seems to be a good trade-off between accuracy and speed on the simulations and the real data. Of course, there may be adversarial case where MaxDepth parameter needs to be increased to yield correct results. However, these adversarial cases seem rare in real data. Although additional speed-up can be achieved by testing multiple subsets at once using amortization techniques such as (Nguyen et al., 2024a), whether there is a general invariance-based polynomial-complexity algorithm remains an open question.

Second, our algorithms assume accurate invariance test and MMSE estimation. The theoretical analysis of the proposed algorithms' performance under imperfect invariance test is challenging since unlike ICP, they are greedy algorithms that do not exhaustively test all subsets. Thus, the performance bounds are problem-specific (dependent on the ground-truth graph) as the invariance test might fail along some greedy optimization path even with perfect MMSE estimation. In practice, MMSE estimation may be inaccurate as well so we need to account for the probability of making a consequential error in ordering the estimated MMSE values. MMSE estimation may be imperfect due to insufficient samples or misspecified model class. Ensuring good MMSE estimation is an empirical exercise that will require careful experimentation. We recommend users to account for the uncertainty in the predicted MMSE via approaches like bootstrapping on the test data (Raschka, 2022) or double ML (Chernozhukov et al., 2018).

**Acknowledgments**

Funding for this project was in part provided by the NIH grants R01AG053949, R01AG064027 and R01AG070988, and the NSF CAREER 1748377 grant.

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
