## A Testing for Invariance

We use the same test for invariance which was used in prior work (Peters et al., 2016; Heinze-Deml et al., 2018; Mogensen et al., 2022). Specifically, the test checks whether the means and variances of the average prediction errors (residuals) across all environments are the same. Algorithm 3 show the pseudo code which checks whether $\mathbf{S}$ is an invariant set. $\mathbf{X}$ and $Y$ are respectively the data of the covariate variables and the target variable combined across all environments. $E$ is a label vector such that each entry in $E$ corresponds to a data sample and samples belonging to the same environment have the same label. Checking for any difference in means is done using the t-test. Checking for any difference in variances is done using the Levene's test (Levene, 1960). The regression model used can be linear or nonlinear. For nonlinear regression model, we used gradient boosted tree (Friedman, 2001). Since high-capacity nonlinear model can overfit, we use cross-validation prediction to avoid overfitting. The number of cross-validation fold is 10 when there are fewer than 500 samples and is 2 otherwise.

---

**Algorithm 3:** $is\_invariant$: invariance testing based on invariant residuals

**Input:** $E, \mathbf{X}, Y, \mathbf{S}, \text{threshold} = 0.05$
**Output:** Is $Y \perp\!\!\!\perp E | \mathbf{S}$?

1   model $= fit(\mathbf{X}[:, \mathbf{S}], Y)$
2   residuals $= Y - predict(\text{model}, \mathbf{X}[:, \mathbf{S}])$
3   pValues $= []$
4   **for** eLabel **do**
5      inGroup $= \text{residuals}[E{==}\text{eLabel}]$
6      outGroup $= \text{residuals}[E{\neq}\text{eLabel}]$
7      pValue1 $= t\_test(\text{inGroup}, \text{outGroup})$
8      pValue2 $= levene\_test(\text{inGroup}, \text{outGroup})$
9      pValues $= \text{pValues} + [2 * \min(\text{pValue1}, \text{pValue2})]$
10   **return** $\min(\text{pValues}) < \text{threshold}$

---

## B Simulation details

Different synthetic datasets are generated by varying the following parameters: (1) number of predictors $d$, (2) number of interventions $N_{\text{int}}$, and (3) the type of intervention. For each set of parameters, the following procedure is repeated 100 times to generate 100 different random graphs.

1. Sample a random acyclic graph $\mathcal{G}$ with $d + 1$ nodes and a pair of nodes in $\mathcal{G}$ is connected with probability $p_{\text{edge}}$ (which is 0.1 for the large dense graph and is $2/N_{\text{int}}$ otherwise).

2. Choose a random node with at least 1 parent to be $Y$.

3. Add a node $E$ with no incoming edges. From of the set $X_1, \ldots, X_d$, pick $N_{\text{int}}$ nodes.

4. If $Y$ is not a descendant of $E$, repeat steps 1–3 until a graph where $Y \in \mathsf{DE}(E)$ is obtained.

$E$ is an environment indicator. A data sample is observational when $E = 0$ and is interventional when $E = 1$ (the children of $E$ may be intervened on).

For each graph, 50 sets of edge coefficients $(\beta_{i \to j})$ are drawn randomly. The coefficients are sampled independently and uniformly from the interval $U((-2, 0.5) \cup (0.5, 2))$. For each set of coefficients, we sample 4 datasets with different sample sizes $n \in \{10^2, 10^3, 10^4, 10^5\}$. Each data sample is generated as follow.

1. Sample $E$ from a Bernoulli distribution with probability $p = 0.5$.

2. Iterate through the nodes in graph in topological order and generate its value:

- If $E = 1$ and the node is a child of $E$, the value depends on the type of intervention.
- Else, the value is generated based on its parents' values using functions $f_j$. Gaussian noise is added afterward.
$$X_j = f_j(\mathsf{PA}(X_j)) + N(0, 1)$$

- There are 4 different types of functions $f$ that were used, namely
  a Linear (Peters et al., 2016): $f_j(\mathsf{PA}(X_j)) = \sum_{i \in \mathsf{PA}(X_j)} \beta_{i \to j} X_i$
  b Nonlinear type 1 (Heinze-Deml et al., 2018): $f_j(\mathsf{PA}(X_j)) = \prod_{i \in \mathsf{PA}(X_j)} \mathsf{sign}(\beta_{i \to j}) g_{ij}(X_i)$
    where $g$ is one of these 4 functions:
    1. $g(x) = x$
    2. $g(x) = \max\{0, x\}$
    3. $g(x) = \mathsf{sign}(x)\sqrt{|x|}$
    4. $g(x) = \sin(2\pi x)$
  b Nonlinear type 2 (Heinze-Deml et al., 2018): $f_j(\mathsf{PA}(X_j)) = \sum_{i \in \mathsf{PA}(X_j)} \beta_{i \to j} g_{ij}(X_i)$ with the same $g$ functions as nonlinear type 1
  b Nonlinear type 3: $f_j(\mathsf{PA}(X_j)) = \sum_{i \in \mathsf{PA}(X_j)} \beta_{i \to j} X_i^2$

We consider 3 types of interventions: (1) perfect interventions, (2) imperfect interventions, and (3) noise interventions (Cooper & Yoo, 1999; Tian & Pearl, 2001; Eberhardt & Scheines, 2007; Peters et al., 2016).

- Perfect interventions: the values of the intervened nodes are set to 1, regardless of the values of their parents.

- Imperfect interventions: the values of the intervened nodes are still the weighted sum of their parents. However, the edge coefficients are modulated by coefficients $\gamma_{i \to j} \sim U(0.0, 0.2)$.

- Noise interventions: the values of the intervened nodes are still the weighted sum of their parents. However, the additive noise is $N(0, 4)$ instead of $N(0, 1)$.

After all data samples are generated, the data are standardized along the causal order to prevent shortcut learning (Reisach et al., 2021).

## C  Additional results

### C.1  Linear Simulations — Perfect Interventions

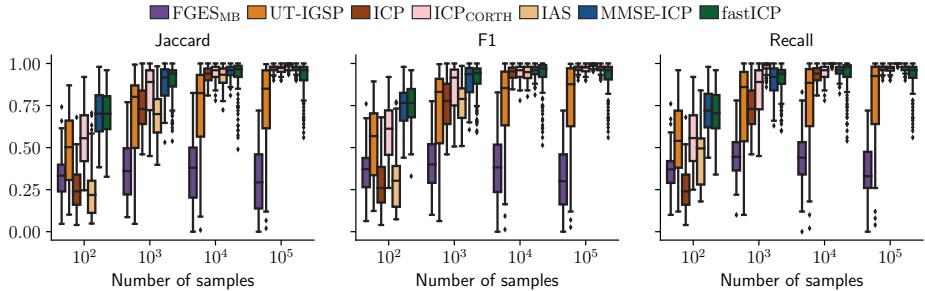

Figure 7: Performance when $N_{\text{int}} = d = 6$ (Table 1, No. 1). Reference set: $\mathsf{PA}(Y)$.

ICP, MMSE-ICP, and fastICP obtain very similar results for large sparse graphs (Figure 11). However, for large dense graph (Figure 12), both MMSE-ICP and fastICP outperform the baselines. Although MMSE-ICP and fastICP achieve similar performance for small graph or large sparse graph in which MMSE-ICP can search exhaustively, when the graph is large and dense, the ability to search through all nodes gives fastICP an edge over MMSE-ICP.

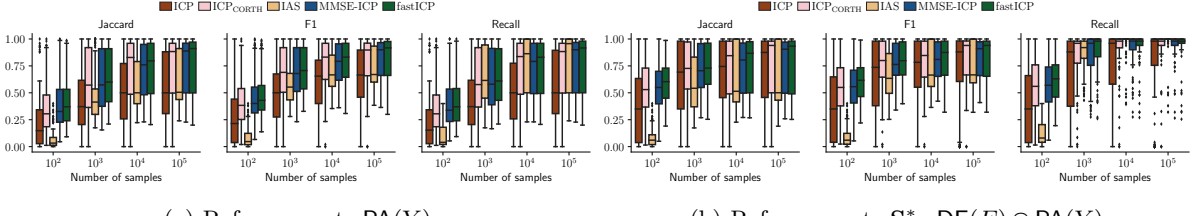

(a) Reference set: PA(Y)          (b) Reference set: $\mathbf{S}^*$=DE(E) ∩ PA(Y)

Figure 8: Performance when $N_{\text{int}} = 1; d = 6$ (Table 1, No. 2)

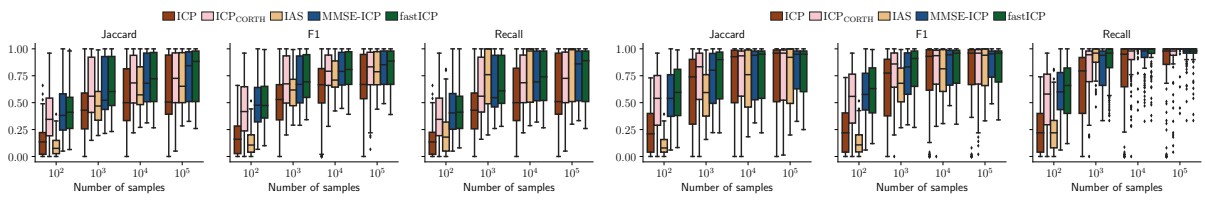

(a) Reference set: PA(Y)          (b) Reference set: $\mathbf{S}^*$=DE(E) ∩ PA(Y)

Figure 9: Performance when $N_{\text{int}} = 2; d = 6$ (Table 1, No. 2)

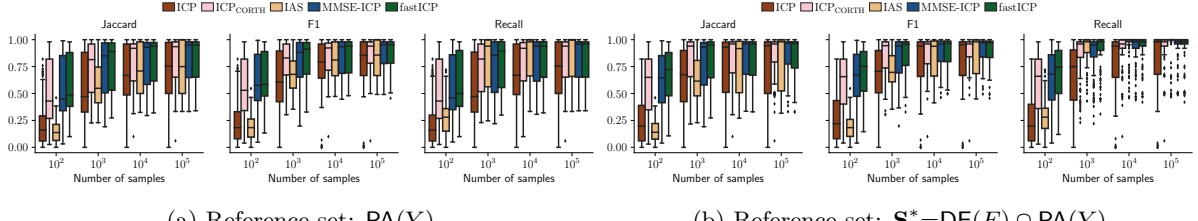

(a) Reference set: PA(Y)          (b) Reference set: $\mathbf{S}^*$=DE(E) ∩ PA(Y)

Figure 10: Performance when $N_{\text{int}} = 3; d = 6$ (Table 1, No. 2)

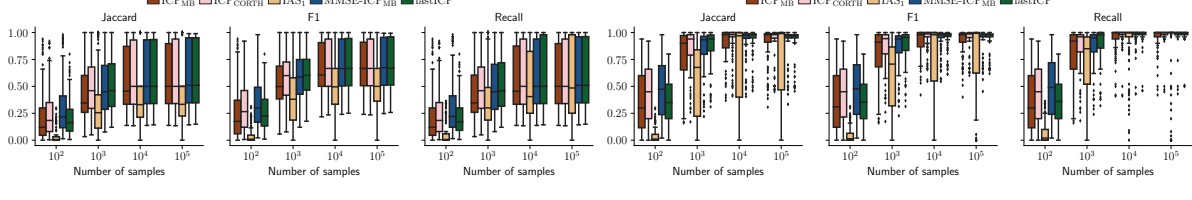

(a) Reference set: PA(Y)          (b) Reference set: $\mathbf{S}^*$=DE(E) ∩ PA(Y)

Figure 11: Performance for large sparse graphs (Table 1, No. 5)

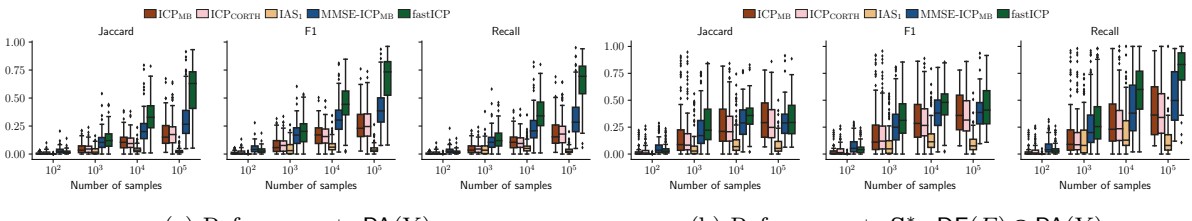

(a) Reference set: PA(Y)          (b) Reference set: $\mathbf{S}^*$=DE(E) ∩ PA(Y)

Figure 12: Performance for large dense graphs (Table 1, No. 6)

## C.2 Linear Simulations — Strong Dependency

Since MMSE used in the proposed algorithms is estimated empirically from data, there may be potential robustness issues with minimizing empirical MMSE. For instance, the proposed algorithms can be sensitive to strong dependency between causal parents and other predictors. To verify the robustness of the proposed algorithms in general and the MMSE estimation in particular, we conduct experiments whereby noisy copies of $X_i$ are created. In particular, after the $X_i$ variables are generated following a noisy linear system with perfect intervention, the copies $X_i'$ of $X_i$ are created by adding Gaussian noise to $X_i$, i.e. $X_i' := X_i + N(0, \epsilon^2)$. We experimented with 2 non-zero values for $\epsilon$: 0.1 and 0.01. The smaller $\epsilon$ is, the stronger the dependency. $\epsilon$ must be non-zero because it is assumed that there is no redundant variables in the system. The algorithms are robust when they do not mistake the noisy copies for the true parents.

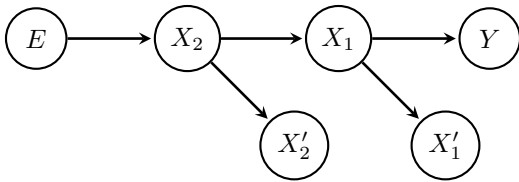

Figure 13: Example of a simulation with strong dependency. $X_i'$ are noisy copies of $X_i$.

Figure 14 and 15 show the results for the two different values of $\epsilon$. The results indicate that the empirical MMSE estimation is quite robust as the proposed algorithms still do relatively well and still outperform the baseline models.

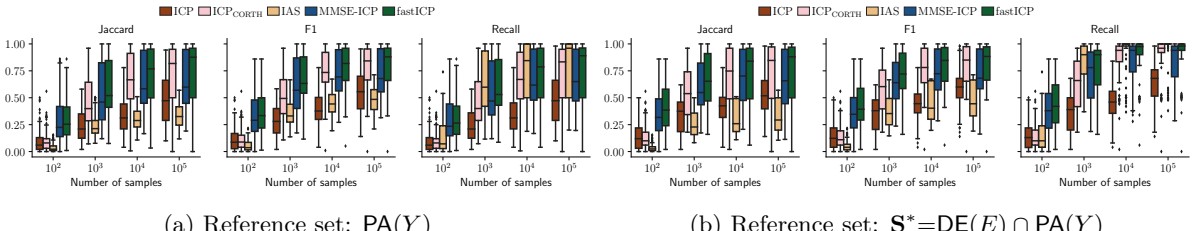

(a) Reference set: $\mathsf{PA}(Y)$      (b) Reference set: $\mathbf{S}^* = \mathsf{DE}(E) \cap \mathsf{PA}(Y)$

Figure 14: Performance when $N_{\text{int}} = 1; d = 6, \epsilon = 0.1$

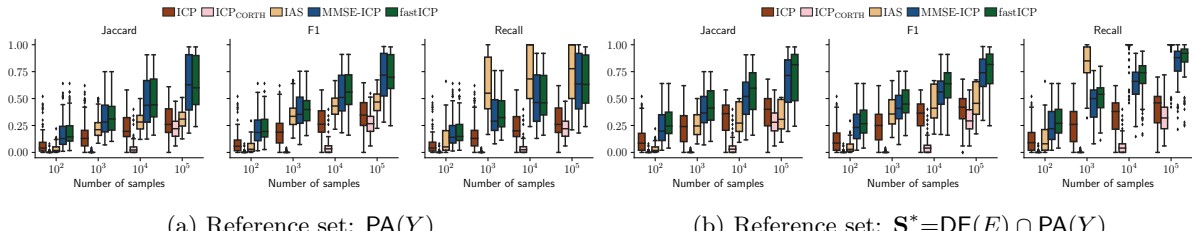

(a) Reference set: $\mathsf{PA}(Y)$      (b) Reference set: $\mathbf{S}^* = \mathsf{DE}(E) \cap \mathsf{PA}(Y)$

Figure 15: Performance when $N_{\text{int}} = 1; d = 6, \epsilon = 0.01$

### C.3 Linear Simulations — Imperfect Interventions

For noise interventions, MMSE-ICP and fastICP achieve similar performance and outperform the baselines in both Jaccard similarity and F1-score (Figure 17). The same trends are observed for imperfect interventions (Figure 16). Although MMSE-ICP and fastICP outperform ICP and IAS for imperfect interventions, there is still a large variance in the Jaccard similarity and F1-score of MMSE-ICP and fastICP. This could be because the approximate test based on residuals of a linear predictor does not have sufficient power to detect this types of changes in mechanisms. Switching to an invariance test with higher detection power might result in better results and lower variance.

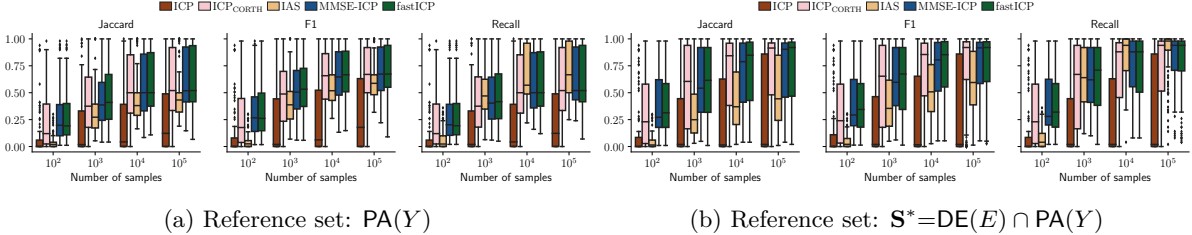

(a) Reference set: $\mathsf{PA}(Y)$        (b) Reference set: $\mathbf{S}^*=\mathsf{DE}(E) \cap \mathsf{PA}(Y)$

Figure 16: Performance under "imperfect" interventions when $d = 6; N_{\text{int}} = 1$ (Table 1, 3rd row)

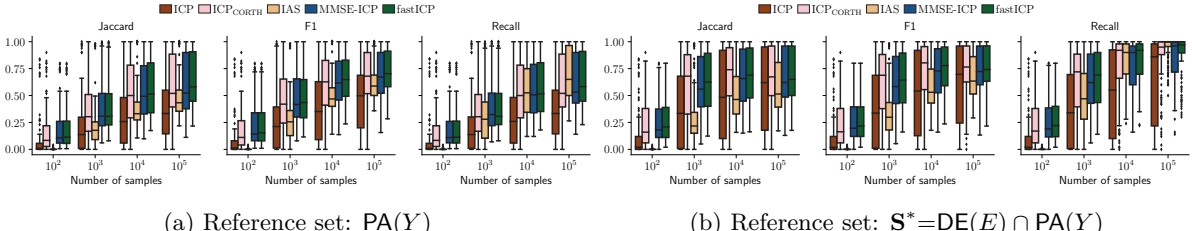

(a) Reference set: $\mathsf{PA}(Y)$        (b) Reference set: $\mathbf{S}^*=\mathsf{DE}(E) \cap \mathsf{PA}(Y)$

Figure 17: Performance under "noise" interventions when $d = 6; N_{\text{int}} = 1$ (Table 1, 4th row)

## C.4 Nonlinear Simulations

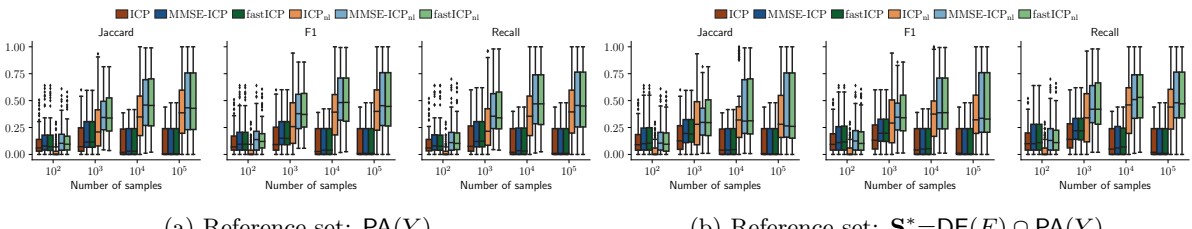

(a) Reference set: $\mathsf{PA}(Y)$        (b) Reference set: $\mathbf{S}^*{=}\mathsf{DE}(E) \cap \mathsf{PA}(Y)$

Figure 18: Nonlinear type 1. $d = 6; N_{\text{int}} = 1$. Same as Figure 5.

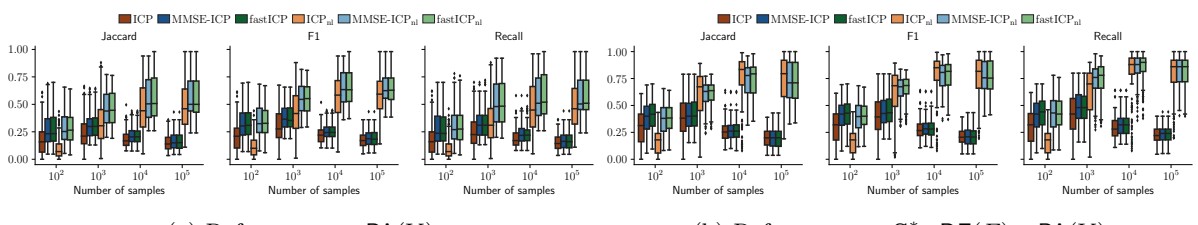

(a) Reference set: $\mathsf{PA}(Y)$        (b) Reference set: $\mathbf{S}^*{=}\mathsf{DE}(E) \cap \mathsf{PA}(Y)$

Figure 19: Nonlinear type 2. $d = 6; N_{\text{int}} = 1$.

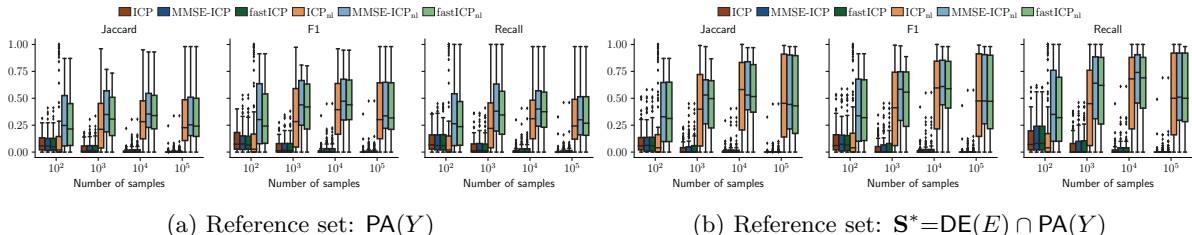

(a) Reference set: $\mathsf{PA}(Y)$        (b) Reference set: $\mathbf{S}^*{=}\mathsf{DE}(E) \cap \mathsf{PA}(Y)$

Figure 20: Nonlinear type 3. $d = 6; N_{\text{int}} = 1$.

## C.5 Real data

We validated the proposed algorithms using the Sachs et al. (2005)'s benchmark which has a well-established causal graph. Since invariance-based algorithms are designed for local causal discovery problem, but the Sachs et al. (2005) benchmark is a global causal discovery, we need to consider each node as a target, one at a time. We adopt the setup proposed by Meinshausen et al. (2016) for this problem. The results on the Sachs et al. (2005) benchmark is shown in Figure 21. For each method, we ran multiple times with different cut-off value $\alpha$. We can see that MMSE-ICP and fastICP are usually better than ICP, having higher true positives at the same level of false negatives.

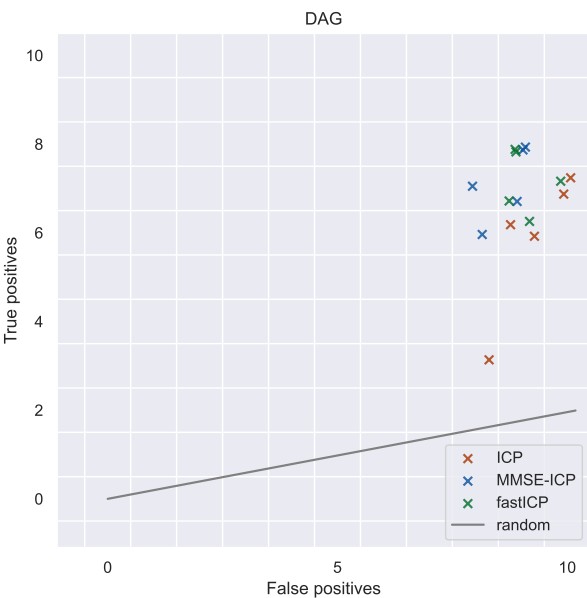

Figure 21: Results of different algorithms on Sachs et al. (2005) benchmark. Each method is ran multiple times with different cut-off value $\alpha$. Solid line is the performance from random guessing.