# OpenReview forum: "Efficient Identification of Direct Causal Parents via Invariance and Minimum Error Testing"
_TMLR — Accepted by TMLR_

### Review · Reviewer_hc2X · 2024-06-07

**Summary Of Contributions:**

This paper considers the problem of discovering the parents of a target variable in a structured causal model. It builds on ICP, proposing the minimum mean squared error of prediction as the criterion (MMSE-ICP). Since ICP requires an exponential number of invariance tests, it also proposes fastICP, which employs a heuristic to prune variables to reduce the computational complexity. Theoretical analysis is done showing that MMSE-ICP is guaranteed to find the causal parents if the invariance tests are accurate and that the heuristic in fastICP is reasonable. Empirical results are shown on synthetic datasets (both linear and nonlinear) and gene expression data. MMSE-ICP and fastICP often outperform ICP and IAS in Jaccard similarity, F1 score, and recall, especially for larger datasets.

**Audience:**

Yes

**Broader Impact Concerns:**

The authors do not include a broader impact statement. I think this is fine because the paper proposes a general purpose causal discovery method, which only has indirect effects by applying it.

**Claims And Evidence:**

Yes

**Requested Changes:**

Critical: Adding analysis of the computation time to the experimental results.

It would strengthen the paper to add some theoretical analysis for the case where the tests of invariance may be incorrect, but I understand that this could be difficult.

**Strengths And Weaknesses:**

### Strengths ###
1. The paper is clearly written.
2. There are theoretical results characterizing when the proposed algorithm is guaranteed to find the parents of the target variable.
3. Experimental results (on both synthetic and real data) indicate that the proposed method is able to find the parents of the target variable more effectively than existing methods in many cases and is competitive in others.

### Weaknesses ###
1. The theoretical results assume that the tests of invariance are accurate, which is fairly strong. The appendix provides some ablations on synthetic data that show that MMSE-ICP and fastICP are robust to increasing the difficulty of these tests, but adding some theory would make the paper stronger.
2. The paper argues that fastICP is able to reduce the computational complexity from a theoretical perspective. It would strengthen the work to show some experimental evidence to this effect, e.g. by recording the wall clock time needed by each method.

---

> ### Author Response · Authors · 2024-07-22
>
> **Add analysis of the computation time to the experimental results.**
>
>
> We recorded the time each method took when the number of nodes ($d$) varies.
> The number of samples, $N$, is fixed at 1000.
> The table below (also included in Appendix B of paper) reports the numbers of seconds elapsed when executing on an AMD EPYC 7642 CPU core (@ 2.3GHz).
> For this benchmark, since the official ICP implementation was in R, we employed a re-implementation in Python for a fair comparison.
> The methods search through the full set of covariates (i.e. no pre-selection of variables using Markov Blanket estimation).
>
> From these runtime values, we confirm that ICP and MMSE-ICP both have exponential complexity in $d$.
> Furthermore, the additional computation overhead of MMSE-ICP as compared to ICP is negligible.
> In contrast, the runtime of fastICP is polynomial (quadratic) in $d$ so fastICP is much more scalable than ICP and MMSE-ICP.
>
>
> | Method   | Language | $d=6$             | $d=9$             | $d=12$             | $d=15$             | $d=18$             | $d=21$            |
> |----------|----------|-------------------|-------------------|--------------------|--------------------|--------------------|-------------------|
> | ICP      | R        | $0.16_{\pm 0.01}$ | $1.78_{\pm 0.06}$ | $17.66_{\pm 0.69}$ | $339.5_{\pm 53.2}$ | $>3600$            | $>3600$           |
> | ICP      | Python   | $0.06_{\pm 0.01}$ | $0.41_{\pm 0.08}$ | $3.28_{\pm 1.06}$  | $30.2_{\pm 10.8}$  | $224.6_{\pm 67.2}$ | $1794_{\pm 593.}$ |
> | MMSE-ICP | Python   | $0.06_{\pm 0.01}$ | $0.42_{\pm 0.08}$ | $3.30_{\pm 1.06}$  | $29.9_{\pm 10.6}$  | $225.5_{\pm 66.1}$ | $1806_{\pm 618.}$ |
> | fastICP  | Python   | $0.03_{\pm 0.00}$ | $0.07_{\pm 0.00}$ | $0.12_{\pm 0.00}$  | $0.20_{\pm 0.01}$  | $0.28_{\pm 0.01}$  | $0.39_{\pm 0.02}$ |
>
> *The average runtime in seconds of different methods for varying $d$.*
>
>
>
> In the revision, we will add this analysis to the results section.
>
>
> **Add some theoretical analysis for the case where the tests of invariance may be incorrect, but I understand that this could be difficult.**
>
> Indeed, the theoretical analysis is challenging, as both MMSE-ICP and fastICP do not exhaustively test all subsets for invariance, unlike ICP and IAS.
>
> As demonstrated in the original ICP paper by~\cite{peters2016causal}, if $H_{0,S}$ is a test of subset $S$ for invariance at level $\alpha$, then ICP's output $\hat{S}$ is a subset of the parents $S^*$ with probability at least $1-\alpha$.
> \begin{equation*}
>     P(\hat{S} \subseteq S^*) =
>     P\bigg[ \bigcap_{S: H_{0,S} \text{ not rejected}} S \subseteq S^* \bigg] \geq
>     P(H_{0,S^*} \text{ not rejected}) \geq 1 - \alpha
> \end{equation*}
>
> The proposed algorithms, MMSE-ICP and FastICP, are greedy versions of ICP and thus this bound does not apply.
> There are two complications. First one is that the invariance tests might fail along the greedy optimization path. Even if we assumed that the MMSE estimation is perfect, we have struggled to derive a useful result without any strong assumptions. We are currently looking into this. We will discuss this point in the revised paper.
>
> Obviously, in practice the MMSE estimation will not be perfect either.
> In this case, we will need to account for the probability of making a consequential error in ordering the estimated MMSE values.
> This can leverage confidence intervals that can be in practice approximated using techniques like bootstrapping.
> We leave this analysis to future work.
>
> These insights will be added to the revision.

---

### Review · Reviewer_qx4X · 2024-06-11

**Summary Of Contributions:**

This work proposes two improvements upon Invariant Causal Prediction (ICP); minimum mean square ICP and fastICP. The main improvements stem from using a specific inequality that allows for principled filtering of nodes where the invariance tests are applied to. The inequality states that the minimum mean squared error (MMSE) of predicting the target variable $Y$ based on specific covariates $X_1 \cup X_2$ is lower than when including just $X_1$. This is a consequence of the fact that the minimum mean squared error considers the best possible estimator, the conditional expectation of the target variable conditioned on the inputs. The authors use this inequality to argue for a selection rule over invariant subsets of the non-descendants of $Y$ (and descendants of the environment $E$) that involves two things; a) use the MMSE to select subsets with the smallest MMSE and b) pick the smallest subset out of those remaining as the parents of $Y$. This is the MMSE-ICP procedure which, despite being generally better than ICP in identifying the parents of $Y$, it still scales exponentially in the number of variables. To combat that, the authors propose fastICP, which first filters the set of all variables based on a specific heuristic from prior work, then applies the invariance test when removing variables from the remaining set one-by-one to create final set of candidates. The parents are then the set of variables that achieves the minimum MMSE. The authors test these methods along with baselines on both simulated and real data demonstrating some improvements upon the baselines.

**Audience:**

Yes

**Claims And Evidence:**

Yes

**Requested Changes:**

This work is generally interesting and, as far as I am aware, novel. The results could also be generally useful for the community. For this reason, I am on the positive side. I only have a few minor remarks / questions:
- The error inequality applies when one can identify the correct estimator and has enough data to do so. It is unclear whether this inequality would still hold with finite data and restricted model classes. While the authors do provide some empirical measurements of the robustness of the results at Appendix D.2, it would be beneficial to discuss how such considerations also affect theoretically the error inequality.
- Corollary 3.2, I do not exactly follow how the first inequality is an immediate corollary of Lemma 3.1; to me it seems to be a direct consequence of the other two inequalities. Perhaps the authors could expand on that part.
- The results presented in the main text are a bit lukewarm; MMSE-ICP and fastICP can yield some improvements on the low data regime but across all experimental settings it seems that the improvements are not statistically significant. Also, it would be good if the authors expand on why fastICP seems to generally be better than MMSE-ICP; from the method discussion, it seems that the main motivation for fastICP was speed at the expense of accuracy.

**Strengths And Weaknesses:**

### Strengths
- The paper is relatively well written and the method intuitive
- The application of MMSE to this setting is, as far as I am aware, novel
- There are performance improvements on real world data

### Weaknesses
- The MMSE inequality is asymptotic and applies to cases where a) one can obtain the actual best regressor and b) there are enough data to do so. Therefore, given the fact that the authors use finite datasets and model classes, it would be good to extend the theoretical analysis with some form of non-asymptotic results.
- There are no clear winners in the results and generally there seems to be one other baseline on par with MMSE-ICP/FastICP.

---

> ### Author Response · Authors · 2024-07-22
>
> **The error inequality applies when one can identify the correct estimator and has enough data to do so. It is unclear whether this inequality would still hold with finite data and restricted model classes. While the authors do provide some empirical measurements of the robustness of the results at Appendix D.2, it would be beneficial to discuss how such considerations also affect theoretically the error inequality.**
>
> The reviewer points out an important weakness in our work that warrants further investigation (see our Response to Reviewer hc2X as well).
> It is important to note that MMSE estimation is imperfect due to insufficient samples or misspecified model class.
> To avoid model misspecification in order to get good MMSE estimation, it can be better to use a more flexible model class, when there are sufficient data.
>  However, choosing a high capacity model class can significantly slow down the algorithm because of long model-fitting time. Furthermore, using flexible models may result in poor estimation due to overfitting, when data is insufficient.
> Hence, ensuring good MMSE estimation is an empirical exercise that will require careful experimentation.
> We recommend users to account for the uncertainty in the predicted MMSE via approaches like bootstrapping on the test data (Raschka, 2022).
>
> We will add this discussion to the updated draft.
>
> 1. Sebastian Raschka. Creating confidence intervals for machine learning classifiers, 2022. URL
> https://sebastianraschka.com/blog/2022/confidence-intervals-for-ml.html
>
> **Corollary 3.2, I do not exactly follow how the first inequality is an immediate corollary of Lemma 3.1; to me it seems to be a direct consequence of the other two inequalities. Perhaps the authors could expand on that part.**
>
> Since $\mathbf{S}\subset\mathsf{ND}(Y)$, it follows from the Causal Markov condition that $Y \mbox{${}\perp\mkern-11mu\perp{}$} \mathbf{S}|\mathsf{PA}(Y)$.
>
> By Lemma 3.1, $\mathsf{MMSE}(\mathsf{PA}(Y)\cup\mathbf{S}){=}\mathsf{MMSE}(\mathsf{PA}(Y))$.
> Furthermore, also by lemma 3.1, $\mathsf{MMSE}(\mathbf{S}\cup\mathsf{PA}(Y)){\leq}\mathsf{MMSE}(\mathbf{S})$.
> Hence, $\mathsf{MMSE}(\mathbf{S}){\geq}\mathsf{MMSE}(\mathsf{PA}(Y))$.
>
> We will clarify this in the updated version.
>
> **The results presented in the main text are a bit lukewarm; MMSE-ICP and fastICP can yield some improvements on the low data regime but across all experimental settings it seems that the improvements are not statistically significant. Also, it would be good if the authors expand on why fastICP seems to generally be better than MMSE-ICP; from the method discussion, it seems that the main motivation for fastICP was speed at the expense of accuracy.**
>
>
> The simulation result may seem underwhelming for the small graphs because when the graphs have few vertices, the number of graphs that are difficult for ICP but not for MMSE-ICP and fastICP are small.
> As the graph size increases (see Figure 4b), we can see that the performance gap between ICP and the proposed methods also grows.
> Furthermore, the result on the real dataset also shows that the proposed methods are much better than ICP.
>
> Regarding the performance difference between MMSE-ICP and fastICP, we may not expect them to always have the same performance because of the order they test subsets is different.
> MMSE-ICP tests subsets in increasing cardinality order, whereas fastICP tests subsets in decreasing cardinality order.
> This order of testing may impact the final accuracy when the test of invariance is imperfect (hence the performance difference).

---

### Review · Reviewer_iEGW · 2024-07-10

**Summary Of Contributions:**

The authors extend existing methods for causal detection to using squared error. This retains the exponential time complexity of ICP, but allows for the method to have identifiability which ICP lacks (without additional restrictions). Additionally, the authors provide a heuristic algorithm that accelerates the original algorithm, MMSE-ICP, by using a pruning size to accelerate removing non-blocking nodes. Both methods perform better than ICP on various simulated settings of causal models, alongside a setting with real data. The faster method also performs better than MMSE-ICP when under compute constraints.

**Audience:**

Yes

**Broader Impact Concerns:**

No direct ethical implications are present from the work, so no broader impacts are left unaddressed.

**Claims And Evidence:**

Yes

**Requested Changes:**

Recommendations:
- I noticed that there was a choice of MaxDepth of 2 in some of the linear settings for fastICP. Is there any reasoning behind this choice? Especially for larger graphs, it seems as though a larger MaxDepth would accelerate fastICP without hindering discovery too greatly. That tradeoff or some mention of anecdotal evidence for why only a MaxDepth of 2 was used would be interesting to have.
- It would be nice to see these methods tested on other real datasets with well-established causal graphs as well.

**Strengths And Weaknesses:**

Strengths:
- Synthetic experiments show, in a variety of settings, that these proposed methods are often more accurate than existing methods for causal detection.
- Notation of common elements from this work and related work (such as ancestors and descendants) is clear in their definitions. These also are all located in a common section, which helped greatly in parsing statements throughout the paper.
- Theorems and lemmas proved defend well that, as long as invariance tests are accurate, MMSE-ICP will find the parents of Y. It's a clear and concise theorem that showcases the validity of this method.

Weaknesses:
- While there are extensive results on simulated settings, there is only a single experiment on real data with expert-knowledge/ground-truth causal graphs.
- In the theory, there's no elaboration on if the invariance tests fail. What comes to mind is a sample complexity result using existing results for invariance tests to create an overall complexity result for MMSE-ICP.

---

> ### Author Response · Authors · 2024-07-22
>
> **Is there any reasoning behind the choice of MaxDepth of 2? Especially for larger graphs, it seems as though a larger MaxDepth would accelerate fastICP without hindering discovery too greatly. That tradeoff or some mention of anecdotal evidence for why only a MaxDepth of 2 was used would be interesting to have.**
>
> The larger the MaxDepth, the closer fastICP is to ICP (i.e. testing exponential number of subsets).
> Hence, setting a smaller MaxDepth will result in greater speed-up.
> In our experiments, MaxDepth of 2 seems to be a good trade-off between accuracy and speed on the simulations and the real data.
> Of course, there may be adversarial case where MaxDepth parameter needs to be increased to yield correct results.
> However, these adversarial cases seem rare in real data.
>
> We will add this discussion to the updated draft.
>
> **Test these methods on other real datasets with well-established causal graphs**
>
> We are not aware of any causal discovery benchmark which has both a large number of variables and a well-established causal graph. The only real dataset with well-established causal graph that we know of is from Sachs et al. (2005). However, this dataset only has 11 variables and we are open to suggestion about which real datasets to use.
>
> Since ICP-based methods are designed for local causal discovery problem, but the Sachs et al. (2005) benchmark is a global causal discovery, we need to consider each node as a target, one at a time. We adopt the setup proposed by Meinshausen et al. (2016) for this problem. For each method, we ran multiple times with different cut-off value $\alpha$. The results on the Sachs et al. (2005) benchmark indicate that MMSE-ICP and fastICP are usually better than ICP, having higher true positives at the same level of false negatives (we are unable to show the figure here since OpenReview does not support showing images).
>
> We will add these results to the updated draft.
>
> 1. Nicolai Meinshausen, Alain Hauser, Joris M Mooij, Jonas Peters, Philip Versteeg, and Peter Buhlmann. Methods for causal inference from gene perturbation experiments and validation. Proceedings of the National Academy of Sciences, 113(27):7361–7368, 2016.
> 2. Karen Sachs, Omar Perez, Dana Pe’er, Douglas A Lauffenburger, and Garry P Nolan. Causal protein-signaling networks derived from multiparameter single-cell data. Science, 308(5721):523–529, 2005. 6

---

> > ### Comment · Reviewer_iEGW · 2024-08-04
> > **Acknoledgement**
> >
> > I acknowledge the authors' rebuttal, and these clarifications/additions are appreciated. As someone who is less familiar with CP, I don't have any other datasets or benchmarks in mind. It's interesting that there aren't more though. This seems like an interesting and common enough problem where there ought to be some datasets developed from physical sciences in the future.

---

### Decision · Action_Editor_sf43 · 2024-08-31

**Recommendation:** Accept as is

**Comment:**

This is a nice paper that improves invariant causal prediction in multiple ways. While the theoretical results make fairly strong assumptions, the overall work innovative and covers some important limitations of the standard existing techniques. All reviewers found that there is substantial value in the work, and I agree with them.

**Audience:**

Yes, work on discovering causal relationships is highly relevant to the TMLR audience.

**Claims And Evidence:**

Yes, the paper provides theoretical and empirical evidence for the set of claims made.